

# Effect of rainfall-induced diabatic heating over southern China on the formation of wintertime haze on the North China Plain

Xiadong An[1], Lifang Sheng[1,2,3], Chun Li[1,3], Wen Chen[4], Yulian Tang[4], Jingliang Huangfu[4]

[1]Department of Marine Meteorology, College of Oceanic and Atmospheric Sciences, Ocean University of China, Qingdao 266100, China
[2]Key Laboratory of South China Sea Meteorological Disaster Prevention and Mitigation of Hainan Province, Haikou 570000, China
[3]Ocean-Atmosphere Interaction and Climate Laboratory, Key Laboratory of Physical Oceanography, Ocean University of China, Qingdao 266100, China
[4]Center for Monsoon System Research, Institute of Atmospheric Physics, Chinese Academy of Sciences, Beijing 100190, China

*Correspondence to*: Lifang Sheng (shenglf@ouc.edu.cn)

**Abstract.** During the winters (December–February) between 1985 and 2015, the North China Plain (NCP) suffered many periods of heavy haze, and these episodes were contemporaneous with extreme rainfall over southern China; i.e., South Rainfall−North Haze events. The formation of such haze events depends on meteorological conditions, which are controlled by the atmospheric circulation associated with rainfall over southern China, but the underlying physical mechanism remains unclear. This study uses observations and model simulations to demonstrate that haze over the NCP is modulated by anomalous anticyclonic circulation caused by the Rossby wave train, in conjunction with the north−south circulation system (NSC), which ascends over southern China, moves north into northern China near 200–250 hPa, and then descends in the study area. Moreover, in response to rainfall heating, southern China is an obvious Rossby wave source, supporting waves along the subtropical westerly jet waveguide and finally strengthening anticyclonic circulation over the NCP. Composite analysis indicates that these changes lead to a stronger descending motion, higher relative humidity, and a weaker northerly wind, which favors the production and accumulation of haze over the NCP. A linear baroclinic model (LBM) simulation reproduced the observed NSC reasonably well and supports the diagnostic analysis. Quasi-geostrophic (QG) vertical pressure velocity (ω) diagnostics were used to quantify the contributions to the NSC made by large-scale adiabatic forcing and diabatic heating. The results indicated that the NSC is induced mainly by diabatic heating related to precipitation over southern China, and the effect of large-scale circulation is negligible. These results provide the basis for a more comprehensive understanding of the mechanisms that drive the formation of haze over the NCP.





# 1 Introduction

Extensive heavy haze on the North China Plain (NCP) has a detrimental effect on both human health and social activities (Chen et al., 2017; Hughes et al., 2018; Lelieveld et al., 2019; Li et al., 2019). These haze events are caused by emissions of pollutants combined with unfavorable meteorological conditions (Yang et al., 2016; Cai et al., 2017; Ding et al., 2017; Zhang et al., 2021). Although emissions play an important role in the generation of haze, numerous studies have suggested that meteorology is also a significant factor in the occurrence of extreme haze events (Quan et al. 2011; Wang et al. 2015;

Gao et al. 2016; Stirnberg et al., 2021). For instance, Dang and Liao (2019) found that large interannual variations in the frequency and intensity of severe winter haze days were driven mainly by changes in meteorology. Chen and Wang (2015) also showed that the occurrence of severe haze events over northern China during the winter generally correlates with meteorological factors. Zhang et al. (2020) provided evidence that the accumulation of pollutants caused by unfavorable meteorological conditions has offset the decreases caused by emissions reductions during the COVID-19 lockdown, leading

to the high aerosol concentrations over Beijing–Tianjin–Hebei that developed between 7 and 14 February 2020. Unfortunately, continued global warming will further increase the incidence of haze days in China by reducing the wind strength (Cai et al., 2017; Xu et al., 2019). Callahan and Mankin (2020) found that climate change will lead to haze-favorable conditions over Beijing becoming more frequent, but that internal variability can generate large uncertainties in these projections. There is little doubt that developing an improved understanding of the factors and mechanisms that causes

haze is one of the greatest challenges facing researchers over the coming decades.

Overall, the role of meteorology in the generation of haze is crucial but uncertain, and may be closely related to the regulation of the large-scale circulation. Large-scale circulation, and the related external forces derived via exciting the teleconnection pattern, regulate meteorological conditions, reduce dispersion, and facilitate the accumulation of haze pollutants (Zhang et al., 2020). Hence, the predictability of wintertime heavy haze over the NCP stems from the underlying

atmospheric conditions. These include, but not limited to: Eurasian snow cover and central Siberian soil moisture (Zhang et al., 2020); the SST anomalies (SSTAs) related to El Niño–Southern Oscillation (ENSO) in the tropical Pacific (Feng et al., 2019; G. Zhang et al., 2019; Yu et al., 2020; Zhang et al., 2020) and the Atlantic Oceans (Wang et al., 2019; Zhang et al., 2020); the Pacific Decadal Oscillation (PDO) (Zhao et al., 2016); the Arctic Oscillation (AO)(Cai et al., 2017; G. Zhang et al., 2019); the preceding Antarctic oscillation (i.e., August−September−October) (Z. Zhang et al., 2019); and the North

Atlantic Oscillation (NAO) pattern (Feng et al., 2019; Li et al., 2021), as well as Arctic sea ice changes (Zou et al., 2020), especially in the Beaufort Sea (Yin et al., 2019a; Li and Yin, 2020) and Chukchi Sea (Yin et al., 2019b).

Recent studies have further documented that heavy haze over the NCP can be attributed to anticyclonic anomalies over northeastern Asia (AANA) caused by westerly-jet wave trains in the middle to upper troposphere over the Eurasian continent (Chen et al., 2019; Wang et al., 2019; An et al., 2020). As a synoptic-scale circulation, these AANA are accompanied by

anomalous southeasterly winds near the surface, as well as a temperature inversion layer and anomalous vertical motion in the surrounding areas, which encourages the development of severe haze (Zhong et al., 2019; An et al., 2020). It is worth



mentioning that An et al. (2020) also found that the atmospheric circulation related to rainfall over southern China further supports the maintenance of heavy haze in the AANA background over the NCP via the local north−south circulation system (NSC; Fig. 1). However, their interpretation was based only on isolated extreme heavy haze events during November and

December 2015 (An et al., 2020). To determine whether this finding was simply a special case, further investigations will be required that are based on greater number of haze events over a longer period. Consequently, this study, we aim to investigate the mechanisms associated with haze formation over the NCP between 1985 and 2015, with an emphasis on the role of circulation related to rainfall-induced diabatic heating in southern China. In addition, given the potential role of circulation related to rainfall over southern China in maintaining heavy haze over the NCP, we will also explore diabatic

heating using a linear baroclinic model (LBM) run under heating forcing in southern China.

The remainder of the paper is organized as follows. The second section describes the datasets and methods used in this work. This is followed by a more extensive section describing the haze events and associated weather patterns. The fourth section introduces the role of diabatic heating caused by rainfall on haze formation over the NCP, which we simulated using the LBM. The paper concludes with a brief summary and discussion of the formation of haze over the NCP.

**2 Data and methods**

**2.1 Data**

We obtained quality-controlled in situ daily rainfall data from 194 stations in China covering the period January 1985 to February 2015 from Chinese Meteorological Administration (CMA, http://www.nmic.cn/) to determine the distribution of rainfall over southern China. Daily visibility data from meteorological stations in the region bounded by 15–55° N, 105–

135° E from January 1985 to December 2015 were also obtained from the CMA. Such observed rainfall (Li and Sun, 2015; Ding and Li, 2017) and visibility (e.g., Liu et al., 2017; An et al., 2020) data have been widely used in previous research into extreme rainfall and haze events. In addition, the daily $PM_{2.5}$ concentration dataset for China for the period 1980–2019 was collated by Y. Yang (2020) using data available at https://zenodo.org/record/4293239#.YJI3J8DiuUn. These daily PM2.5 concentrations were in excellent agreement with ground measurements, with a coefficient of determination of 0.95 and mean

relative error of 12% (Li et al., 2021).

The atmospheric reanalysis data including geopotential, zonal and meridional wind, relative humidity, air temperature, and vertical velocity, were obtained from the National Centers for Environmental Prediction−National Center for Atmospheric Research (NCEP−NCAR) NCEP−DOE AMIP-II Reanalysis (R-2) dataset (Kanamitsu et al., 1996), provided by the NOAA/OAR/ESRL PSD, Boulder, Colorado, USA (http://www.esrl.noaa.gov/psd/). The data used in this study cover a 31-

year period from 1985 to 2015, with a 1-day resolution, a horizontal spatial resolution of 2.5° longitude × 2.5° latitude, and vertical levels from 1000 to 100 hPa.





## 2.2 Methods

In accordance with the standards set by the CMA (2010), we defined haze as a day on which the daily mean visibility and relative humidity were less than 10 km and 80%, respectively, and when no rain, snow, or sand and dust storms occurred.
This identification method has been applied to the forecast of haze-fog by the CMA and is particularly useful for studying haze (e.g., Liu et al., 2017; An et al., 2020).

To analyze the distribution of heating associated with rainfall, we calculated the atmospheric apparent heat source ($Q_1$) according to the equation obtained by Yanai et al. (1973):

$$Q_1 = c_p \frac{\partial T}{\partial t} - c_p(\omega\sigma - \mathbf{V}\cdot\nabla T), \tag{1}$$

where $c_p$ denotes the specific heat at constant pressure, $T$ is the air temperature, $t$ is the time, $\omega$ is the vertical pressure velocity, the static stability $\sigma = (RT/c_p p) - (\partial T/\partial p)$, R is the gas constant, p is the pressure, $\mathbf{V}$ is the horizontal wind vector, $\nabla$ is the horizontal gradient operator, $L$ is the latent heat of condensation, and $q$ is the specific humidity. Here, $Q_1$ represents the total diabatic heating (including radiation, latent heating, and surface heat flux) and the subgrid-scale heat flux convergences (Yanai et al., 1973).

According to Nie et al. (2020), the quasigeostrophic (QG) vertical pressure velocity ($\omega$) diagnostics is can be used to decompose $\omega$ in extreme precipitation into one part ($\omega_D$) due to large-scale adiabatic forcing (F) and another part ($\omega_Q$) due to diabatic heating (Q). The QG$\omega$ equation reads:

$$\left(\partial_{pp} + \frac{\sigma}{f^2}\nabla^2\right)\omega = -\frac{1}{f}\partial_p Adv_\zeta - \frac{R}{pf^2}\nabla^2 Adv_T - \frac{R}{pf^2}\nabla^2 Q, \tag{2}$$

where $\sigma = -\frac{RT}{p}\partial_p \ln\theta$ is the dry static stability, and $f$ is the Coriolis parameter. the terms $Adv_\zeta = -\vec{V_g}\cdot\nabla\zeta$ and $Adv_T = -\vec{V_g}\cdot\nabla T$ are
the horizontal advection of geostrophic absolute vorticity ($\zeta$) and temperature ($T$), respectively, by the geostrophic winds.

To explore the propagation of anomalous Rossby waves along the subtropical westerly jet waveguide in the Northern Hemisphere, the horizontal stationary wave activity flux can be calculated, as follows (Takaya and Nakamura, 2001):

$$\mathbf{W} = \frac{p\cos\varphi}{2|\mathbf{U}|} \cdot \begin{pmatrix} \frac{U}{a^2\cos^2\phi}\left[\left(\frac{\partial\psi'}{\partial\lambda}\right)^2 - \psi'\frac{\partial^2\psi'}{\partial\lambda^2}\right] + \frac{V}{a^2\cos\phi}\left[\frac{\partial\psi'}{\partial\lambda}\frac{\partial\psi'}{\partial\phi} - \psi'\frac{\partial^2\psi'}{\partial\lambda\partial\phi}\right] \\ \frac{U}{a^2\cos\phi}\left[\frac{\partial\psi'}{\partial\lambda}\frac{\partial\psi'}{\partial\phi} - \psi'\frac{\partial^2\psi'}{\partial\lambda\partial\phi}\right] + \frac{V}{a^2}\left[\left(\frac{\partial\psi'}{\partial\phi}\right)^2 - \psi'\frac{\partial^2\psi'}{\partial\phi^2}\right] \end{pmatrix}, \tag{3}$$

where $\mathbf{W}$ is the wave activity flux with unit of $m^2\ s^2$, $\psi$ ($= \Phi/f$) is the geostrophic stream function, $\Phi$ (gpm) is the
geopotential height, and $f$ ($= 2\Omega\sin\phi$) is the Coriolis parameter. $\mathbf{U}$ ($= (U, V)^T$; $m\ s^{-2}$) is the basic flow. We used the daily reanalysis data; i.e., the daily zonal wind, meridional wind, and anomalous geopotential height (for the stream function) to calculate the vector $\mathbf{W}$.



We calculated the linearized Rossby wave source according to Sardeshmukh and Hoskins (1988), which can be expressed as follows:

$$S = -\nabla_H \cdot \left\{ \mathbf{u}_\chi' (f + \bar{\zeta}) \right\} - \nabla_H \cdot \left\{ \mathbf{u}_\chi' \zeta' \right\}. \tag{4}$$

Here, $\mathbf{u} = (u, v)$ denotes the horizontal wind velocity, $\nabla_H$ is the horizontal gradient, and the subscript $\chi(\psi)$ represents the divergent (rotational) component. Overbars indicate the climatological mean and primes signify anomalies.

The linear baroclinic model (hereafter, LBM; Watanabe and Kimoto, 2000), a simple dry model, has been widely used to examine the steady linear response to idealized diabatic heating (Lu and Lin 2009; Sample and Xie, 2010; Xu et al., 2020; Hu et al., 2020). This model consists of basic equations linearized with respect to the mean state of the December (0)– February (1) (DJF) climatology from the NCEP−DOE reanalysis for 1981–2010. The version used here has a horizontal resolution of T42 (roughly equivalent to 2.8°) and 20 vertical sigma levels. Using the time integration methods, the LBM was run up to 30 days, and the variable (i.e., zonal wind, meridional wind and vertical velocity) on the last day (i.e., day 30) was taken as the steady response to the prescribed diabatic heating over southern China.

## 3 Weather patterns related to heavy haze events on the NCP

Ding and Li (2017) investigated 30 extreme winter rainfall events associated with the subtropical westerly jet waveguide over southern China. From 22 rainfall events between 1985 and 2015 and the visibility observation data analysed in this study, we found 13 periods when the NCP experienced a pollution episode at the same time that heavy rainfall occurred in southern China (Table 1; Fig. 2). We took these 13 episodes as representative examples of the occurrence of haze over the NCP when that coincided with periods of heavy rainfall over southern China and defined them as South Rainfall−North Haze (hereafter, SR−NH) events. When rain fell over southern China, the probability of a haze event over the NCP during our research period of 1985−2015 was 59.09%. Details of these 13 SR−NH events are shown in Fig. 2 and Table 1. For the events studied here, there was evident precipitation in southern China, with unfixed rainfall areas (Ding and Li, 2017). At the same time, significant haze was trapped on the NCP, with poor visibility (<10 km) over large areas, which is similar to the conditions described by An et al. (2020). In particular, more intense and widespread haze events occurred over the NCP on 22–24 December 2007, 1–5 January 1992, and 12–17 January 2012. A natural question that arises: what kind of atmospheric circulation regulates the weather phenomena associated with SR−NH events?

Figure 3 shows the composite anomalous air temperature at 1000 hPa, the absolute values of relative humidity at 925 hPa, and the anomalous wind vectors at 1000 hPa for the 13 SR−NH events. The positive air temperature anomaly over the NCP, which is caused by warmer and humid airflow from the low-level western North Pacific and transported by the easterly wind anomaly (Fig. 3a and c), creates favorable moisture conditions for the hygroscopic growth of haze particles, and further deteriorates haze in the NCP (Ding et al., 2017). Additionally, relative humidity over China shows a triode pattern, with low values over the NCP (≤80%), meaning that these haze cases are haze rather than haze−fog (An et al., 2020). However, the





relative humidity over southern China is close to 100%, and this is the result of the southwesterly airflow along the coast of
southern China and the rainfall (Figs 2, 3b, c and 4b). The anomalous easterlies on the eastern side of the NCP not only
transport warm and moist air that promotes the development of haze over the NCP, but they also weaken the East Asian
winter monsoon (EAWM) (An et al., 2020), which is not conducive to the horizontal diffusion of haze (Fig. 3c).

To illustrate the reasons for the above changes in the meteorological factors, we next consider large-scale circulation. The
composite map of the meridional wind anomaly reveals a substantial wave train with alternate positive and negative
meridional wind anomalies at 200 hPa over the mid-latitudes of the Northern Hemisphere (Fig. 4a). This circulation pattern
is suggestive of a Rossby wave train emanating from the North Atlantic (Li and Sun, 2015; Ding and Li, 2017; An et al.,
2020; Li et al., 2020; Huang et al., 2020). According to the Rossby wave theory (Hoskins and ambrizzi, 1993), the
subtropical westerly jet, as a waveguide, allows the Rossby wave to arrive at the NCP (Fig. 4a). The wave activity fluxes
show this wave train extending eastwards from the North Atlantic to eastern Europe, then southeastwards to Arabia, the
Tibetan Plateau, and finally to the NCP (Fig. 4a), indicating that atmospheric circulation over the NCP is regulated by this
wave train. As a result, there is a prominent anticyclonic anomaly over the NCP (Fig. 4b), in which agrees with the analysis
of An et al. (2020). At 500 hPa, the anomalies are similar to the negative phase of the EU, although they are shifted south
and west of the canonical position of the three centers of the EU pattern (Wallace and Gutzler, 1981) (Fig. 4b). This negative
EU-like pattern is not conducive to the development of the EAWM (An et al., 2020), and this further encourages the
development of heavy haze over the NCP. In addition, the AANA related to EU-like pattern also supports haze development
over the NCP via the anomalous descending motion and southerly wind along the east coast of China (Figs 3c and 4b). In
addition, the anomalous southwesterly airflow along the coast of southern China transports large amounts water vapor to this
area, and is one of the conditions that leads to the heavy rainfall over southern China (Li and Sun, 2015; Ding and Li, 2017).

Previous studies have demonstrated that the convergence and divergence anomalies in the upper troposphere can be regarded
as an effective Rossby wave source for the stationary Rossby wave (Hoskins and Ambrizzi, 1993; Branstator, 2002;
Watanabe, 2004; Chen et al., 2020). The strong divergence causes the rainfall in southern China, and is located at 200 hPa
(Li and Sun, 2015; Ding and Li, 2017; An et al., 2020), but does is also act as the Rossby wave source to strengthen the
Rossby wave along the subtropical westerly jet waveguide? The composite map of the Rossby wave source for SR−NH
events reveals a significant negative Rossby wave source in the upper troposphere over southern China around 25°–30° N,
110°–120° E (Fig. 5), where positive precipitation and strong ascending motion anomalies are located (Figs 2 and 6b).
Therefore, the divergence anomalies in the upper troposphere (Fig. 7a) associated with positive precipitation and anomalous
ascending motion (Figs 2 and 6a, b) play a crucial role in the formation of the strong negative Rossby wave source over
southern China, which excites the Rossby wave that propagates to the downstream regions. This means that there is a
positive feedback process involving subtropical westerly jet waveguide-rainfall over southern China. The rainfall over
southern China is largely caused by the subtropical westerly jet waveguide (Li and Sun, 2015; Ding and Li, 2017; Li et al.,
2020), which would in turn would strengthen vertical motion and high-level divergence by an intensification of latent heat


release, generating a Rossby wave source, and hence strengthen the wave train along the subtropical westerly jet. This process reinforces the Rossby wave that originated from the North Atlantic. Wang et al. (2019) and An et al. (2020) found that such waves can account for haze over the NCP. The ascending motion over southern China that is related to the diabatic

heating caused by the rainfall will be examined further below.

## 4 Possible physical mechanisms driving SR−NH events: the role of rainfall-induced diabatic heating

In the previous section, we presented a diagnostic analysis of the atmospheric circulation during periods of haze over the NCP. We now focus on the influence of circulation related to diabatic heating on haze formation over the NCP. The composite sections show when precipitation occurred in southern China, and that it was accompanied by a strong vertical

ascending motion (Fig. 6a, b). At the same time, the NCP was controlled by an obvious descending motion (Fig. 6a, c), which was related to the ascending motion over southern China and forms the NSC together with the ascending motion in southern China (Fig. 6a). The descending motion over the NCP develops with the AANA as shown in Fig. 4b. The divergent wind, as well as the velocity potential, also confirmed that there was strong divergence near 200 hPa over southern China and strong convergence in the northern China (Fig. 7a). The latitude−height cross section of the stream function and relative

vorticity shows that there was strong convergence in the middle- to upper levels of the troposphere, but weak divergence (strong convergence) at lower (higher) levels over the NCP (Fig. 7b), and this was responsible for the descending motion and not conducive to the diffusion of haze. The centers of the convergence and divergence over southern China were the opposite to those over the NCP (Fig. 7b), which is the typical circulation pattern that accompanied rainfall (Li and Sun, 2015). The ascending motion over southern China is not only related to the subtropical westerly jet waveguide (Ding and Li, 2017), but

may also related to the diabatic heating caused by the rainfall.

The diabatic heating in the atmosphere caused by precipitation may intensify the local ascending motion (Wang et al., 2019; Xu et. al., 2020). To estimate the diabatic heating produced by rainfall extremes in southern China, we calculated $Q_1$ quantities using Eq. 3 as described above. Figure 8 shows the composite $Q_1$ during the period of heavy rainfall in southern China. An obviously positive $Q_1$ appears in southern China when rainfall occurs, with the maximum value of $Q_1$ near 26° N,

115° E (Fig. 9a). Short-wave radiation from the sun rarely reaches the lower troposphere during rainfall because it is blocked by the clouds; therefore, a positive value for $Q_1$ indicates the rainfall process may release large amounts of heat. To investigate the vertical distribution of $Q_1$, we calculated the average value of $Q_1$ value between 1000 and 100 hPa over the domain 20°–30° N, 110°–120° E. Figure 9b shows that the maximum value of $Q_1$ (2.6 K day$^{-1}$) occurred at 500 hPa (Fig. 9b). The results shown in Fig. 9a also confirm that the rainfall acts as a heat source. Analyzing the distribution of $Q_1$ may

help to identify the specific heating processes that are occurring in the atmosphere (Yanai et al., 1973). The above analysis corroborates that the rainfall process does release a lot of heat, which may further encourage ascending motion.



To further validate the above-mentioned rainfall–heating circulation, a numerical experiment was conducted. The results outlined above indicate that the NSC seems to be closely associated with the diabatic heating over southern China. Using the LBM, we performed a numerical simulation to complement the observational results and test the plausibility of the proposed

haze–rainfall link. This numerical experiment simulated the atmospheric response to heat forcing induced by heavy rainfall in southern China (Fig. 10a). The experiment was run with diabatic heating centered over southern China (26° N, 115° E), which essentially matched the maximum value and variance of $Q_1$, and the heavy rainfall located as shown in Fig. 8a. The maximum heating, with an amplitude of 2 K day$^{-1}$, was set at 500 hPa (Fig. 8b). Figure 9 illustrates the 300-hPa wind response to the diabatic heating over southern China. An obvious divergence in the wind vectors occurs over eastern China.

The whole layer is divergent, which means that diabatic heating is conducive to upward motion, and the airflow diffuses northwards at 200 hPa.

The NSC simulated by the LBM bears a striking resemblance to the observed spatial pattern of the NSC (Figs 5 and 10), both of which are broadly similar to the results obtained by An et al. 2020. As depicted in Fig. 11b, the upper-level negative and low-level positive relative vorticity over southern China (20°−30° N) indicates that the atmosphere is strongly

baroclinic, with an anticyclonic circulation in the upper levels and a cyclonic circulation at lower levels over southern China. The direct product of this circulation is strong ascending motion over southern China. Meanwhile, upper-level cyclonic circulation and low-level anticyclonic circulation over the NCP supports descending motion there, which is conductive to haze. In addition, at 200 hPa, the positive relative vorticity located over the NCP reinforces the AANA (Fig. 11a). In summary, this LBM experiment further confirms that the circulation related to the rainfall over southern China plays an

obvious assisting role in maintaining haze over the NCP.

From our earlier discussion, it is clear that the diabatic heat released by rainfall over southern China has a significant effect on the ascending motion, except for large-scale circulation background; i.e., the subtropical westerly jet waveguide and south branch trough suggested by Li and Sun (2015). However, a crucial issue that remains to be addressed is which contributes more to the vertical movement? In reality, in a moist atmosphere, the vertical motion depends not only on

dynamic forcing by large-scale perturbations, but also on the driving by the latent heating released from convection (Nie et al., 2020). To quantify the contribution of vertical motion due to rainfall heating, we calculated the components of the ω equation as described in Eq. 4. First, we find that negative (positive) ω values occur mainly over southern China (the NCP), which indicates that southern China is controlled by ascending (descending) motion (Fig. 12). Both the ascending motion over southern China and the descending motion over the NCP are strongest at 500 hPa (Fig. 12). The top panel in Fig. 12

presents the diabatic heating term, which is in agreement with that of ω, especially in the mid−high levels of the troposphere (i.e., 300 and 500 hPa), whereas the dry dynamic forcing term (middle panel in Fig. 12) clearly differs from ω, and its values in the lower level of troposphere (below 700 hPa) are very weak and unlikely to be responsible for enhancing vertical motion (Fig. 12). This means that the strengthened ascending motion caused by the subtropical westerly jet waveguide enhances the production of rainfall over southern China, resulting in increased diabatic heating, which in turn reinforces the ascending



motion and hence forms a positive feedback that links the diabatic heating to the ascending motion when there is a plentiful supply of moisture. The above analysis again confirms that diabatic heating related to rainfall plays a crucial role in the ascending motion over southern China, which in turn maintains the NSC and finally aggravates the haze pollution over the NCP.

**5 Discussion and conclusions**

Our study investigated the mechanisms associated with production over the NCP related to rainfall heating over southern China. We found that the appearance of haze over the NCP is the product of the circulation associated with rainfall over southern China in conjunction with two wave trains along the subtropical westerly jet and polar front jet waveguides. The extreme rainfall events over southern China analyzed in this study were caused mainly by the subtropical westerly jet waveguide (Li and Sun, 2015; Ding and Li, 2017), and the diabatic heat related to it induces the secondary circulation

(referred to as NSC in this paper). On the one hand, this heating strengthens the ascending motion over southern China and the descending motion over the NCP (Fig. 13). On the other hand, this heating stimulates the wave train as the Rossby wave source and strengthens the background subtropical westerly jet wave train. These changes eventually lead to the strengthening of the anomalous anticyclone in northeast Asia, resulting in the strong descending movement, weak northerly wind, and warm and moisture-laden flow over the NCP. As a consequence, heavy haze is maintained over the NCP.

When the AANA over the NCP moves a little to the east (Fig. 14b), heavy rainfall also occurs in southern China and triggers a similar NSC (not shown), but there is then no haze (i.e., visibility was >10 km) over the NCP (i.e., a SR−noNH event). This implies that the AANA is one of the important factors in the occurrence of severe haze events over the NCP (Fig. 4), and this agrees with the results of Chen et al. (2019) and An et al. (2020). In addition, $PM_{2.5}$ concentrations were lower (most areas below 120 ug m$^{-3}$) in the first seven days during nine SR−noNH events (Fig. 15). This implies that emissions are the

source of haze over the NCP, and meteorological factors worsen haze (e.g., Wang et al., 2015; Stirnberg et al., 2021).

In addition to rainfall over southern China associated with the subtropical westerly jet waveguide, rainfall is also associated with ENSO (Ma et al., 2018). Therefore, ENSO may also be, except for the westerly jet waveguide, another potential factor in haze production over the NCP and rainfall over southern China. For instance, there has been rainfall over southern China and haze over the NCP during El Niño years (i.e., 1988, 1992, 2007, and 2015). The present study does not consider the role

of ENSO with respect to rainfall over southern China and haze over the NCP, but this will be the focus of future work. The complex interconnections among atmospheric systems make it difficult to determine whether any single factor is the dominant control on haze production over the NCP, and it is possible that the synergistic effects of many influencing factors may play a more important role in the occurrence of haze. As mentioned in the introduction, there are many meteorological factors that play an important role in haze production in northern China. Previous in-depth studies have considered various

aspects of the formation mechanisms of haze in northern China; i.e., ENSO (Yu et al., 2020; Zhang et al., 2020), the AO (Cai



et al., 2017), and Arctic sea ice (Yin et al., 2019a, b; Li and Yin, 2020). However, whether such factors have a synergistic effect on haze production (e.g., Arctic sea ice and the tropical ocean) has not been considered here but would be a worthwhile focus for a future study.

*Data availability.* The visibility observational data are available at the China Meteorological Administration (http://data.cma.cn/, CAM, 2017), the reanalysis dataset is available at NCEP/NCAR (https://www.esrl.noaa.gov/psd/data/gridded/, NCEP/NCAR, 2020), and the daily PM$_{2.5}$ concentration dataset for China for the period 1980–2019 is available at https://zenodo.org/record/4293239#.YJI3J8DiuUn (last access: 12 April 2021).

*Author contribution.* XA, LS and LC designed the experiments and carried them out. XD and YL developed the model code
and performed the simulations. XA downloaded and analysed the reanalysis data and prepared all the figures. XD prepared the manuscript with contributions from all co-authors. LS, WC, JHuangfu revised the manuscript.

*Competing interests.* The authors declare that they have no conflict of interest.

*Acknowledgement.* This research was supported by the National Natural Science Foundation of China (grant no. 41975008) and Fundamental Research Funds for the Central Universities (grant no. 201961004). All the authors are very grateful to the
China Meteorological Administration (http://data.cma.cn/, last access: 12 November 2017) and NCEP/NCAR (https://www. esrl.noaa.gov/psd/data/gridded/, last access: 20 December 2020) for data used in this study.

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





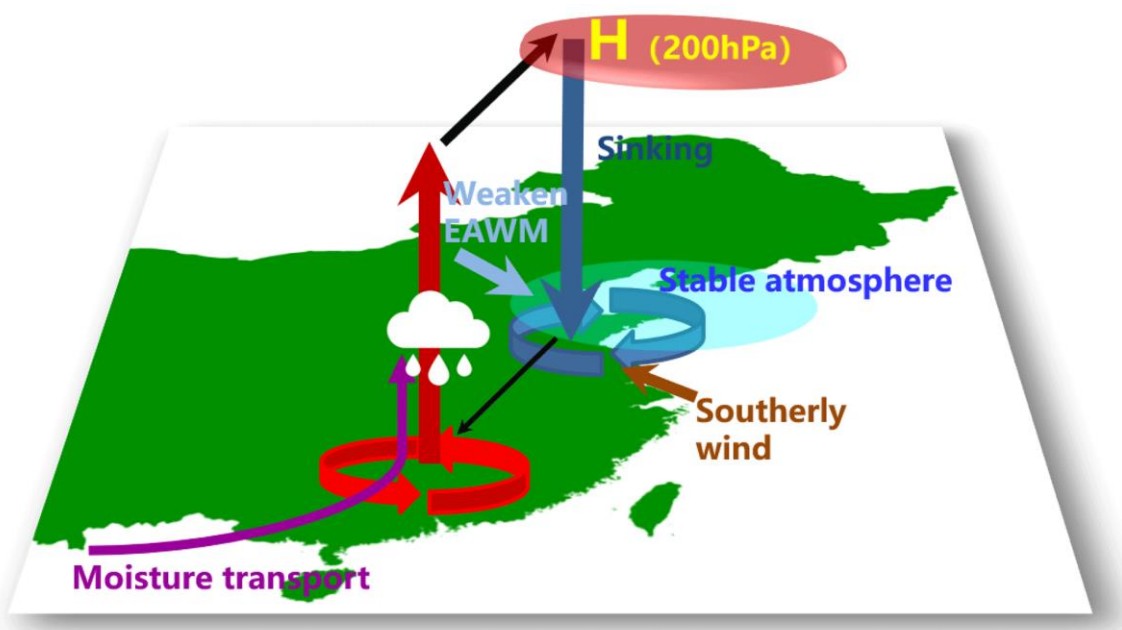

**Figure 1: Schematic illustration of the NSC. The red (blue) circular arrow indicates the anomalous cyclone (anticyclone), the red (blue) vertical arrow indicates ascending (descending) motion, the thick black arrow indicates northward movement in the upper-level troposphere, H indicates the anticyclonic anomaly, and the white cloud indicates rainfall (see An et al. (2020) for caption details).**





**Figure 2: Daily mean visibility (shading, km d⁻¹) over the NCP (30°−40° N, 112°−120° E) and precipitation (shading, mm d⁻¹) in southern China (10°−20° N, 110°−120° E) for each SR−NH event. Panels (a)−(n) represent events 1−13, respectively, as described in Table 1.**

**Figure 3: Composite (a) anomalous air temperature at 1000 hPa, (b) absolute values of relative humidity at 925 hPa, and (c) anomalous wind (arrows) and wind speed (shading, m s⁻¹) at 1000 hPa for the 13 SR−NH events. The white dotted region indicates areas at the 99% confidence level based on the two-tailed Student's _t_ test. The black box indicates the NCP (30°–40.5° N, 112°–121.5° E) and the gray area is the Tibetan Plateau.**





**Figure 4: Composite map of (a) 200 hPa meridional wind anomalies (shading, unit: m s⁻¹, dashed and solid green contours for −6 and 7 m s⁻¹, respectively), zonal wind (contours, unit: m s⁻¹), and wave activity flux (vectors, unit: m⁻² s⁻²), (b) 500 hPa geopotential height anomalies (shading, unit: gpm, dashed and solid green contours for −40 and 20 gpm, respectively) and 850 hPa wind (vectors). White dotted regions indicate areas at the 99% confidence level based on the two-tailed Student's *t* test.**






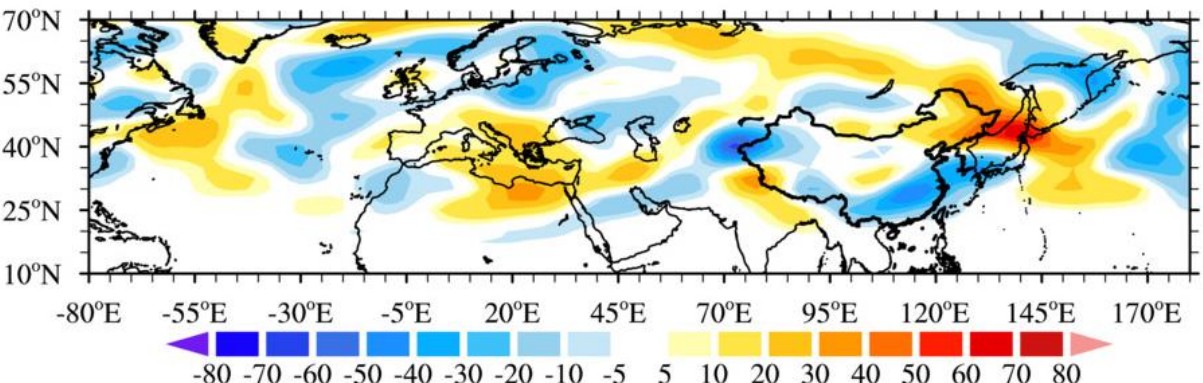

**Figure 5: Composite map of the Rossby wave source (shading, unit: $s^{-2}$) at 200 hPa for the 13 SR−NH events. According to Eq. 4, the variables should be composited first, and then used to calculate the Rossby wave source, hence there is no *t*-test here.**


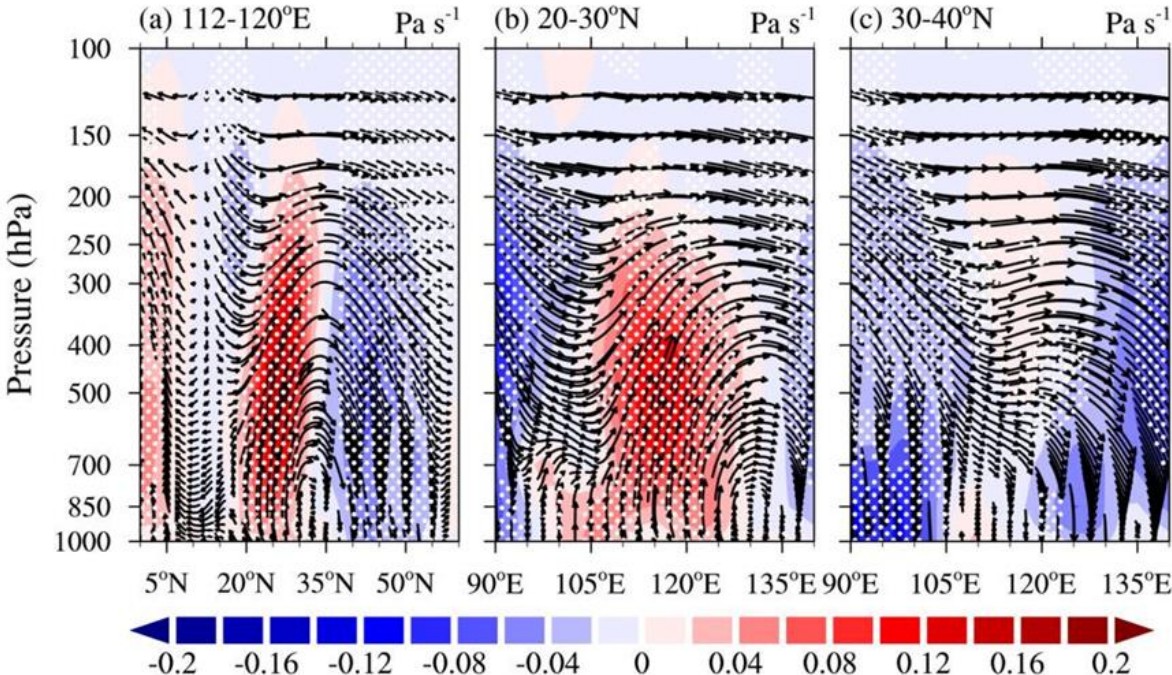

**Figure 6: Composite sections of absolute values for the 13 SR–NH events: latitude–height sections of (a) vertical velocity (shading, unit: Pa s⁻¹) and wind vectors (v and ω) averaged over 112°–120° E, and longitude–height cross sections of (b) vertical velocity (shading, unit: Pa s⁻¹) and wind vectors (u and ω) averaged over 20°–30° N and (c) 30°–40° N. White dotted regions indicate areas at the 99% confidence level based on the two-tailed Student's _t_ test.**

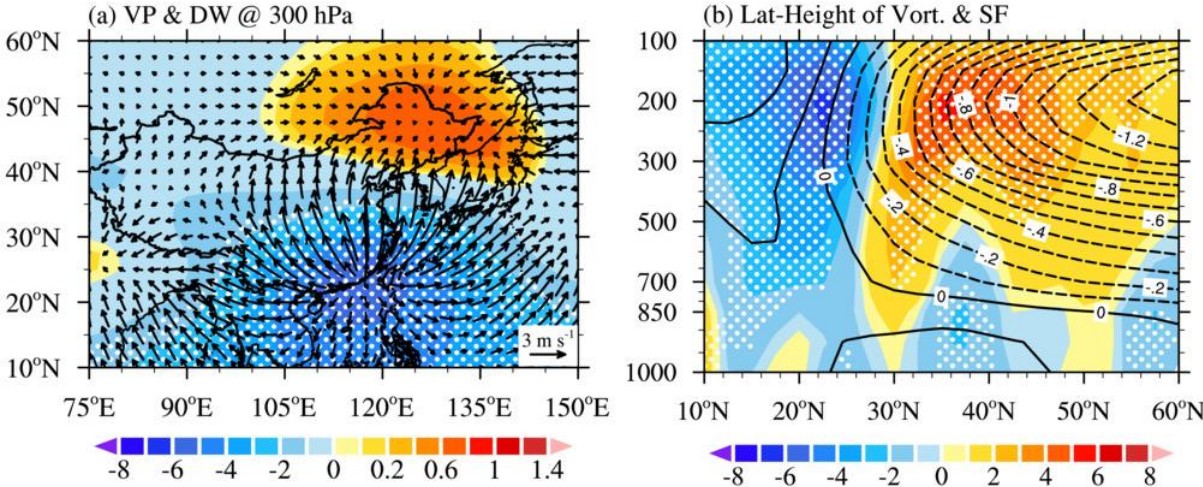

**Figure 7: Composite (a) divergent wind (arrows) and velocity potential (shading, unit: $10^6$ m$^2$ s$^{-1}$) at 200 hPa, (b) latitude–height cross section of the stream function (contours, unit: $10^8$ m$^2$ s$^{-1}$) and relative vorticity (shading, unit: $1.5 \times 10^{-5}$ m$^2$ s$^{-1}$) averaged over 112°–120° E, for the 13 SR–NH events. White dotted regions indicate areas at the 95% confidence level based on the two-tailed Student's $t$ test.**


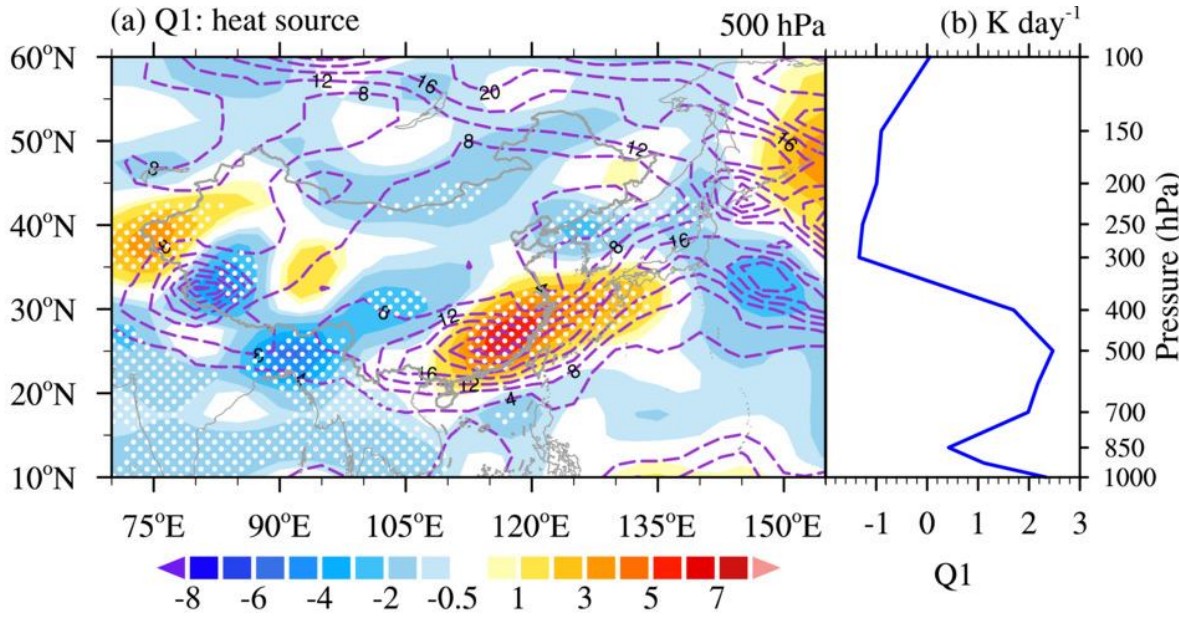

**Figure 8: Composite (a) $Q_1$ (shading, unit: K day$^{-1}$) and variance (dashed contours, unit: %) at 500 hPa for the 13 SR–NH events, and (b) for the composite vertical profile of the average $Q_1$ value over the domain 20°–30° N, 110°–120° E. White dotted regions indicate areas at the 95% confidence level based on the two-tailed Student's *t* test.**

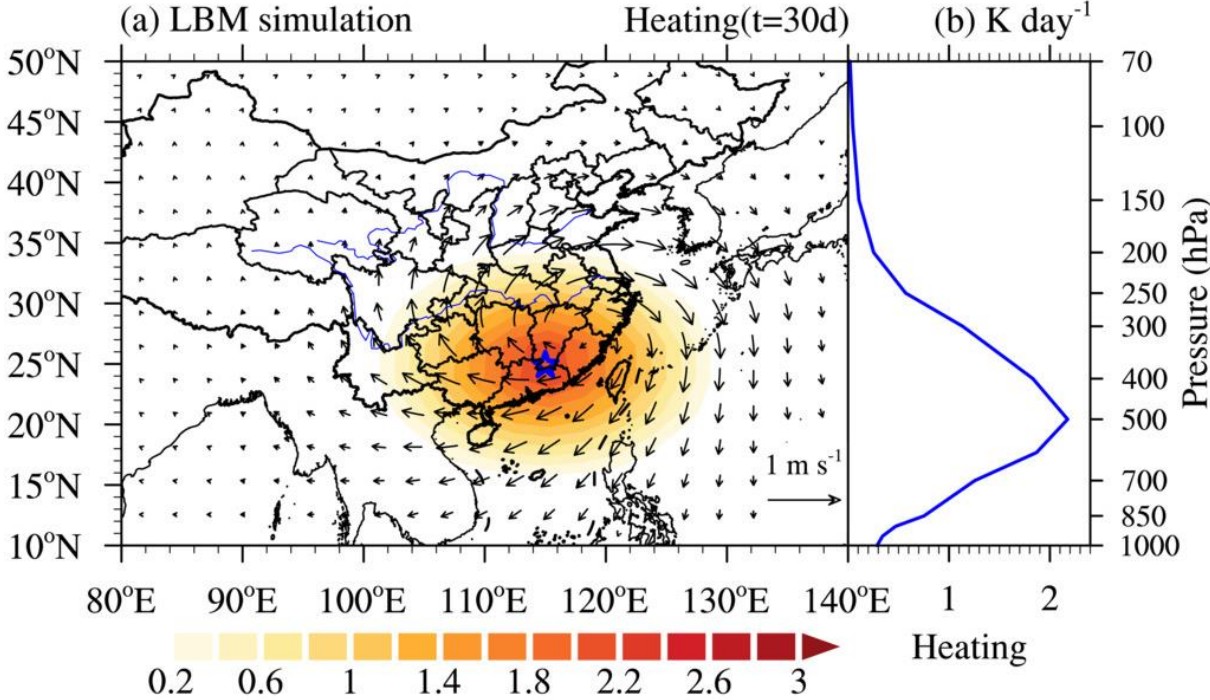

**Figure 9: (a) Heat forcing at 500 hPa (shading, unit: K day$^{-1}$) and the steady response of wind at 300 hPa (vectors, unit: m s$^{-1}$) and**
**(b) profile of the heat forcing at location marked in by the blue star in Fig. 9a.**





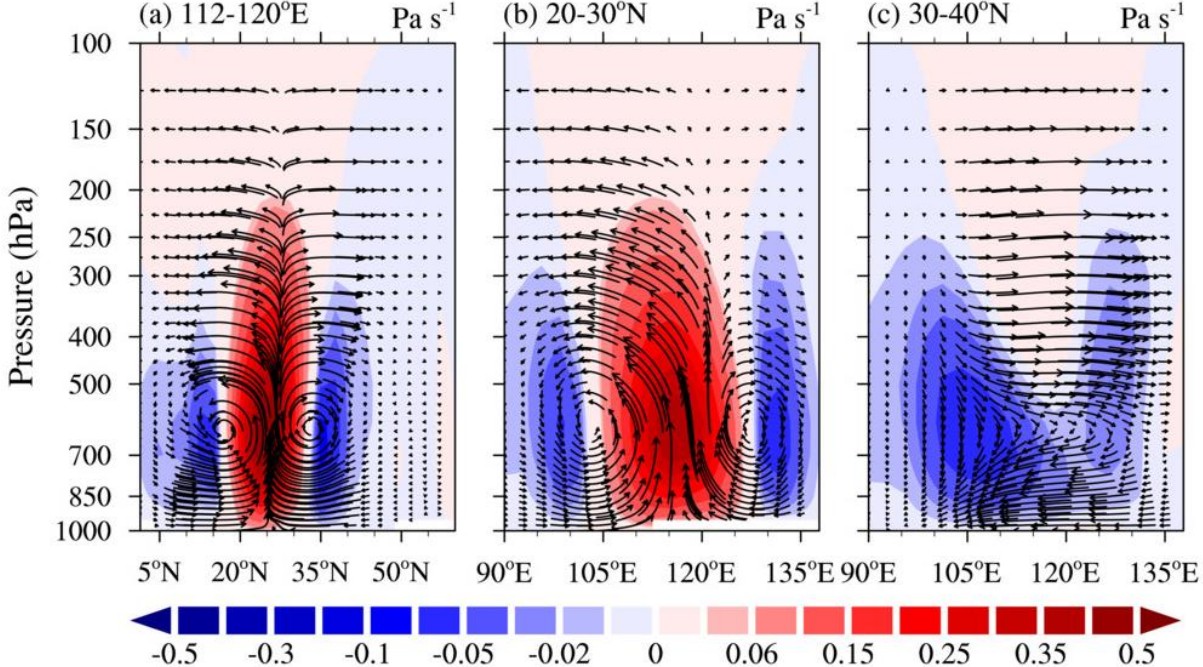

Figure 10: As Fig. 5, except for the results from the LBM. For clarity, ω is multiplied by −20.





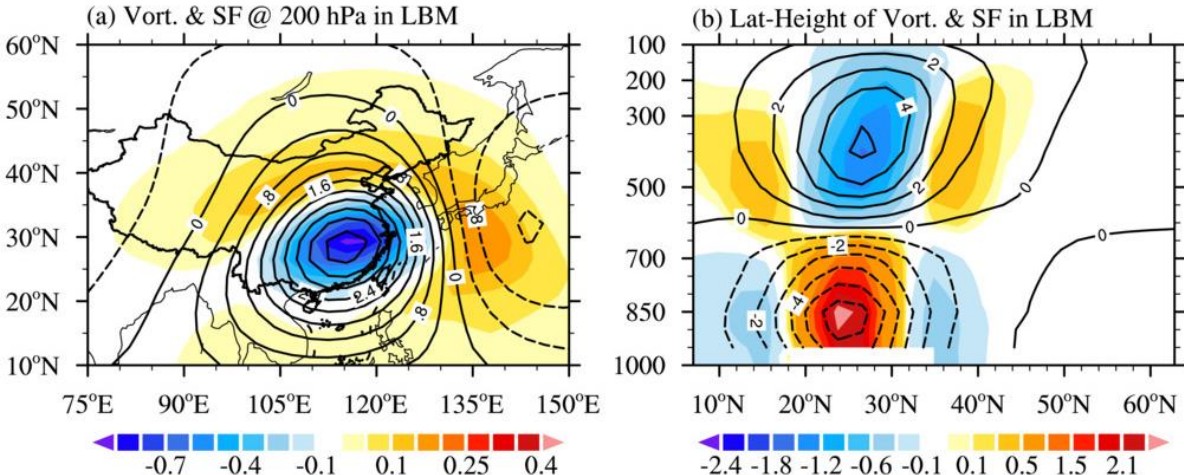


**Figure 11: (a) Horizontal distribution of stream function (contours, unit: $10^6$ m$^2$ s$^{-1}$) and relative vorticity (shading, unit: $10^{-6}$ m$^2$ s$^{-1}$) at 200 hPa in the LBM. (b) Latitude─height cross section of the stream function (contour, unit: $10^6$ m$^2$ s$^{-1}$) and relative vorticity (shading, unit: $10^{-6}$ m$^2$ s$^{-1}$) averaged over 112°–120° E in the LBM.**





**Figure 12: Composite QG decomposition for the 13 SR-NH events. Each column shows a different level. From top to bottom, the rows show the diabatic heating term (- $\frac{R}{pf^2}\nabla^2 Q$), the dry forcing term ($F$ =- $\frac{1}{f}\partial_p Adv_\zeta$ - $\frac{R}{pf^2}\nabla^2 Adv_T$), where $Q$ is calculated using the Eq. 1), and $\omega$ (reanalysis data), respectively. The unit of $\omega$ is Pa s⁻¹. The unit of the $F$ and Q terms is Pa⁻¹ s⁻¹. For ease of viewing, and to reduce the differences in scale, the $F$, Q terms, and $\omega$ were multiplied by $4 \times 10^7$, $0.8 \times 10^7$, and 4, respectively. White dotted regions indicate areas at the 95% confidence level based on the two-tailed Student's t test.**







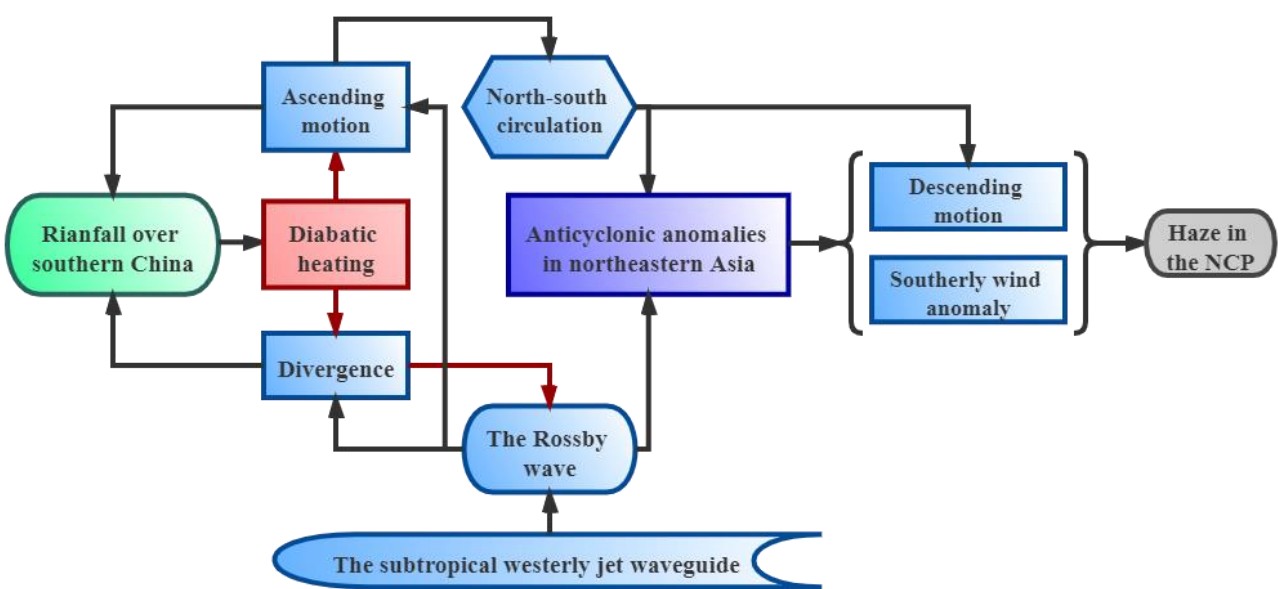

**Figure 13: Schematic representation of the impact of heavy rainfall over southern China on haze production over the NCP.**


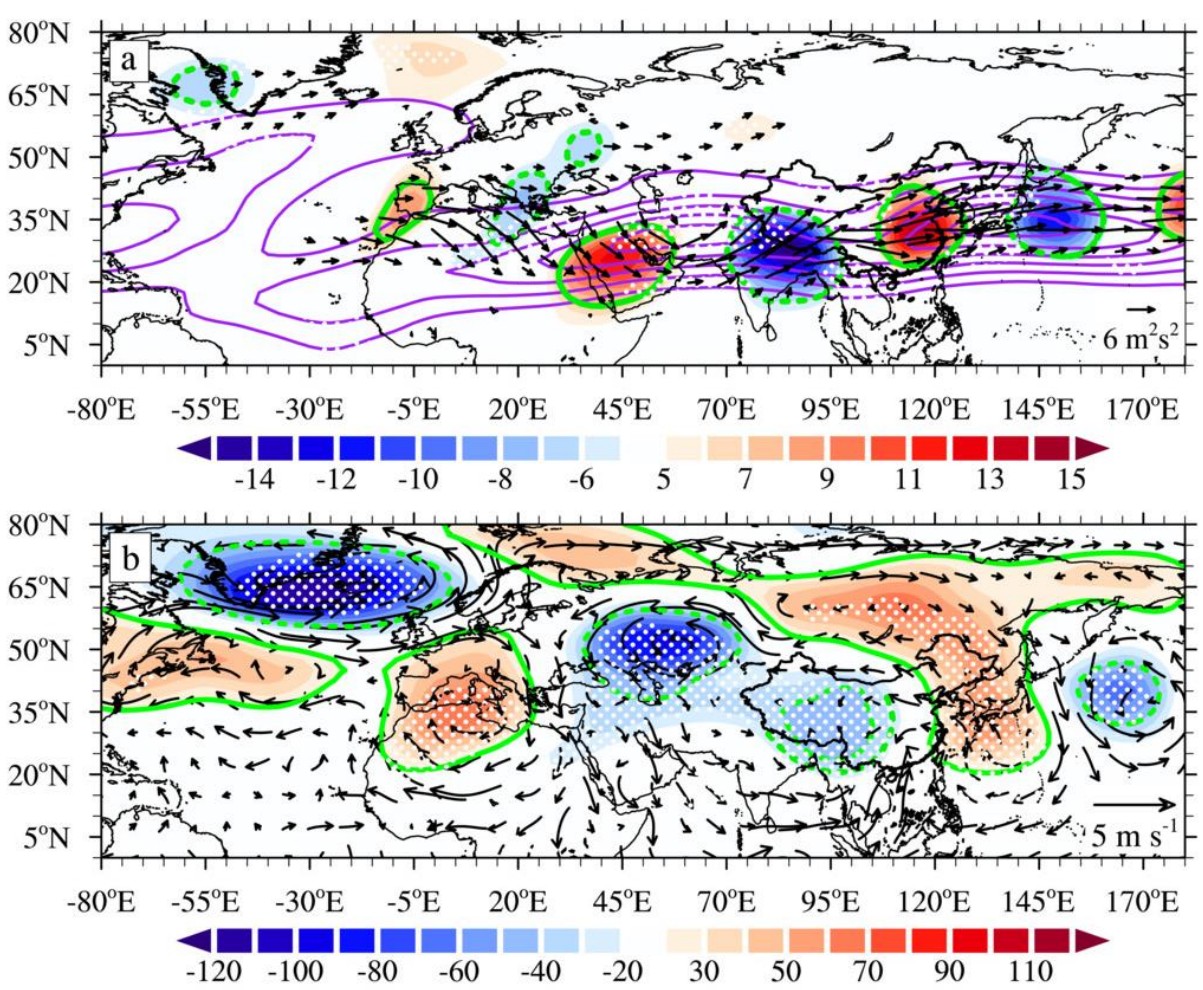

**Figure 14:** As Fig. 4, but for the 9 SR−noNH events in Table 2.





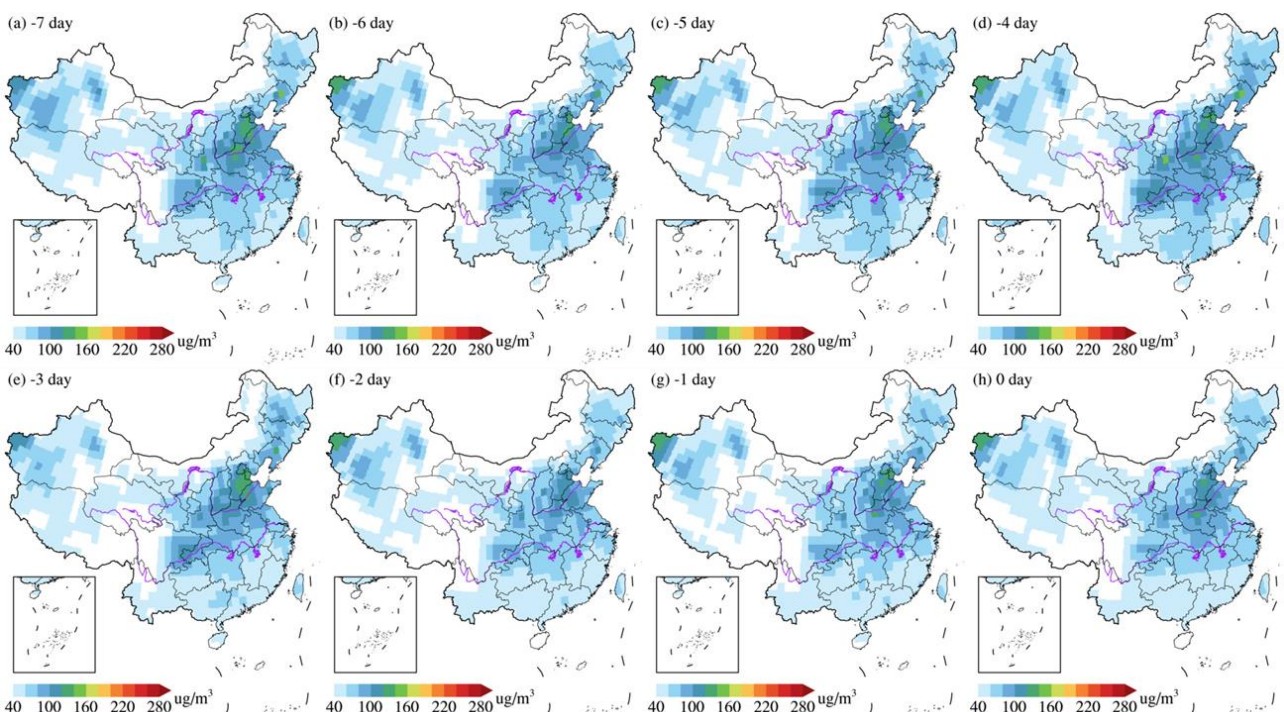

**Figure 15: Composite PM$_{2.5}$ concentration for the 9 SR−noNH events in Table 2 at (a) −7, (b) −6, (c) −5, (d) −4, (e) −3, (f) −2, (g) −1, and (h) 0 days.**



**Table 1. Start and end dates, and duration, of each SR−NH event.**

| No. | Start and end dates | Duration (days) | No. | Start and end dates | Duration (days) |
|-----|---------------------|-----------------|-----|---------------------|-----------------|
| 1 | 4–8 Feb 1985 | 5 | 8 | 13–15 Dec 2006 | 3 |
| 2 | 29–31 Dec 1988 | 3 | 9 | 22–24 Dec 2007 | 3 |
| 3 | 4–8 Jan 1989 | 5 | 10 | 25 Feb – 6 Mar 2009 | 10 |
| 4 | 1–5 Jan 1992 | 5 | 11 | 12–17 Jan 2012 | 6 |
| 5 | 7–12 Dec 1994 | 6 | 12 | 13–17 Dec 2013 | 5 |
| 6 | 7–11 Jan 1998 | 5 | 13 | 8–13 Jan 2015 | 6 |
| 7 | 17–20 Dec 2002 | 4 | | | |




**Table 2. Start and end dates, and duration, of each SR−noNH event.**

| No. | Start and end dates | Duration (days) | No. | Start and end dates | Duration (days) |
| --- | --- | --- | --- | --- | --- |
| 1 | 7–10 Feb 1982 | 5 | 6 | 24–26 Jan 2001 | 3 |
| 2 | 3–8 Jan 1983 | 3 | 7 | 24–28 Feb 2001 | 3 |
| 3 | 27 Feb– 3 Mar 1983 | 5 | 8 | 25 Jan – 2 Feb 2008 | 10 |
| 4 | 15–19 Feb 1985 | 5 | 9 | 21–23 Jan 2010 | 6 |
| 5 | 16–18 Feb 1998 | 6 | | | |