# Peer review of "Effect of rainfall-induced diabatic heating over southern China on the formation of wintertime haze on the North China Plain"

_Atmospheric Chemistry and Physics, 2021_

## Author Comment (AC1)

Dear reviewer #1,

We appreciate you for carefully reviewing our manuscript and providing the valuable suggestions to improve our paper. We have carefully read all comments and revised the manuscript as suggested. The following are our responses to all comments point by point. The italicized sentences are all comments, and the other sentences are the author's responses. The green sentences and words are the specific revisions. We also marked all relevant changes in the manuscript in the same way.

Anonymous Referee #1

*Review of Effect of rainfall-induced diabatic heating over southern China on the formation of wintertime haze on the North China Plain (acp-2021-402):*

*An and the co-authors found when extreme rain fell over southern China, the probability of a haze event over the NCP during our research period of 1985−2015 was 59.09%. Further analysis revealed that the secondary circulation, associated with rainfall over southern China, in conjunction with Rossby wave trains along the waveguides may apt to the haze pollution over the NCP. This is important for understanding of haze pollutions over China. Although this manuscript fit into the scope of ACP, there are still several concerns must be addressed before publication.*

*Major comments:*

1. *The major concern is the small samples and half (13) and half (9) category. I suggest the authors to carefully illustrate the necessity of research and the robust of your revealed relationship. 50.9% is not a high probability.*

**Response:** Thanks for your suggestion. We are very sorry that we made mistakes when we counted the numbers of rainfall events. There are 19 rainfall events from 1985 to 2015, among which 13 cases accompanied by heavy haze in the NCP, namely, the proportion of which was about 68.42%

of the total. The new proportion does not change our conclusions. We have revised this proportion in the manuscript.

30    In this study, it should be noted that we selected haze cases in the NCP based on heavy rainfall cases in southern China selected by Ding and Li (2017), which means that atmospheric backgrounds of haze in the NCP might be same as that of rainfall in southern China. It is helpful to investigate the possible mechanism of the role of rainfall on haze under the similar background. The above method of selecting cases limits partly the numbers of haze cases.

35    In addition, to further check a robust relationship between haze in the NCP and rainfall in southern China, we calculated the EOF of $PM_{2.5}$ concentration and precipitation (Fig. 1Sa). The second EOF mode of $PM_{2.5}$ concentration, with a variance of 15.5% of total, shows a dipole pattern of $PM_{2.5}$ concentration in eastern China. The first EOF mode of precipitation represents pattern of rainfall in southern China with a variance of 56.6% of total (Fig. 1Sb). The second EOF of $PM_{2.5}$ and the first

40    EOF of precipitation together represent the pattern of South rainfall−North pollution. The correlation coefficient between the PC index of these two EOF mode is −0.64 (with a t-test level of 0.01), meaning that rainfall in southern China and air pollution in the NCP tend to occur simultaneously. The correlation coefficient between the interannual component of $PM_{2.5}$ and Precipitation is −0.70 (with t-test level of 0.01).

45    To further illustrate the necessity of research, we have limited the background of atmospheric circulation for the 68.42% probability in the manuscript.

[Figure]

**Figure 1S:** The spatial distribution of EOF mode for 1979−2018 NDJ (a) PM$_{2.5}$ concentration (EOF2) and (b) precipitation (EOF1) within domain 10°N−60°N, 100°E−140°E. (c) The standardized principal component of EOF2 of PM$_{2.5}$ concentration (PC2) and EOF1 of precipitation (PC1). The variance explained by EOF2 of PM$_{2.5}$ concentration and EOF1 of precipitation are 15.5% and 56.6%, respectively. The North-test of these two EOF modes are true. The gray region denotes the Tibetan Plateau (An et al., 2021).

**Line 66−67:** "… that are based on greater number of haze events with the similar background of atmospheric circulation as rainfall in southern China …"

**Line 69:** "… southern China under the similar background of atmospheric circulation."

**Line 134:** "From 19 rainfall events …"

**Line 139:** "… our research period of 1985−2015 was 68.42% under the similar background of atmospheric circulation."

**Line 503:** "… but for the 6 SR−noNH events in Table 2."

**Line 505:** "… for the 6 SR−noNH events …"

**Table 2:**

Table 2. Start and end dates, and duration, of each SR−noNH event.

| No. | Start and end dates | Duration (days) | No. | Start and end dates | Duration (days) |
|-----|---------------------|-----------------|-----|---------------------|-----------------|
| 1 | 15–19 Feb 1985 | 5 | 4 | 24–28 Feb 2001 | 3 |
| 2 | 16–18 Feb 1998 | 6 | 5 | 25 Jan – 2 Feb 2008 | 10 |
| 3 | 24–26 Jan 2001 | 3 | 6 | 21–23 Jan 2010 | 6 |

*2. Possibly, to discuss the differences between SR-NH and SR−noNH is a helpful way. That is, to answer why there is no haze pollution in North China form the perspective of physical mechanisms.*

**Response:** Thank you very much. In this study, we have discussed the differences of atmospheric circulation between SR−NH and SR−noNH in section of discussion and conclusions. In the mid and upper troposphere, the differences between these two weathers seem to be weak except that the 500-hPa northeast Asian anomalous anticyclone is more westerly and the subtropical westerly jet wave train seems to be stronger in SR-NH events (Fig. 4 and Fig. 14). In addition, the atmospheric boundary layer height in SR−NH events is significantly lower than that in SR−noNH events (Fig. 8S of response for reviewer#2), which means that the descending motion related to the local north−south vertical circulation might be conducive to the formation of haze in the NCP. The above results indicate that rainfall in southern China is one of the potential factors affecting haze in the NCP (68.42% in this study), but not all rainfall events in southern China will affect haze in the NCP. For why there is no haze in the NCP in some rainfall events, it might be related to other factors, such as the East Asian winter wind (Liu et al., 2017; Zhao et al., 2018) or stratospheric process

(Huang et al., 2021). Our current study really can't answer this question, and it's a good topic for further study in the future. We have stressed this deficiency in the section of discussion to further discuss for more scientists in the future.

**Line 271−272:** "A comprehensive understanding of why there is no haze in the NCP despite some rainfall in southern China (6 SR−noNH events) needs to be added in the future."

3. *It is difficult to say the anomalous large-scale atmospheric circulations over North China were simulated by the extreme rainfall. Possibly, it is a dipole pattern of atmospheric anomalies, which simultaneously influenced the haze in North China and rainfall in South China. The diabatic heating might be an attendant phenomenon. Thus, more robust evidences are required to make your arguments stand.*

**Response:** Yes, your consideration is right. The anomalous large-scale atmospheric circulations over North China were mainly related to the teleconnection wave trains such the EU pattern or the subtropical westerly wave train (Li et al., 2019; An et al., 2020; Zou et al., 2020). In addition, the dipole pattern of atmospheric anomalies in eastern China indeed can simultaneously influence haze in North China and rainfall in South China. In current research, however, we highlight the role of the local north−south vertical circulation related to rainfall in southern China on haze in the NCP. The strengthening of this local north−south vertical circulation might be related to ascending motion in southern China (Fig. 6 of the manuscript). While the ascending motion in southern China is related to the diabatic heating of rainfall (Figs. 11, 12 and 13). In addition, more shallow atmospheric boundary layer height in the NCP due to an anomalous descending motion also supports the role of the local north−south vertical circulation on haze in the NCP (Fig. 8S). We have revised some of the expressions in the manuscript to make it easier for readers to understand our intentions according to your suggestion.

**Line 16:** "… which are related to …"

**Line 18:** "… anomalous anticyclonic circulation caused by the two Rossby wave trains, …"

*4. Related to the above comment, I suggest the authors the re-plot the schematic diagram or provided more evidence to enhance the story line.*

**Response:** Thanks for your suggestion. We have replotted the schematic diagram in the manuscript.

[Figure]

**Figure 13:** Schematic representation of the impact of heavy rainfall over southern China on haze production over the NCP.

*5. Figure 9, 11 and related texts: I did not find anticyclonic responses over North China.*

**Response:** We are sorry for this confusion. In this study, we highlight that diabatic heating related to rainfall in southern China supports the local north−south vertical circulation by strengthening anomalous upward motion in southern China (Figs 10, 11 and 12). From current results, there is no anticyclonic responses over North China from the LBM model. However, in Fig. 5, we find there is a strong Rossby wave source in southern China, supporting the propagation of the Rossby wave along the subtropical westerly jet, which might be a result of the positive feedback between the upward motion and anomalous precipitation in southern China. An et al. (2020) also found that almost all of the Rossby wave energy dispersing from the upstream along subtropical westerly jet waveguide was converged within southern China (Fig. 2S). It is worth noting that there is the Rossby wave energy in the north of South China dispersing to North China, which might be the result of the positive feedback between an anomalous upward motion and precipitation in southern China (Fig. 2S). In the LBM model, there is no the similar positive feedback. In addition, unlike the Gill-Matsuno response of the tropical atmosphere (Matsuno, 1966; Gill, 1980), there does not seem

to be a classical theory to support that diabatic heating on subtropical land will excite an (a) anticyclone (cyclones) on its north side. Therefore, we don't find anticyclonic responses over North China in Figs 9 and 11.

[Figure]

**Figure 2S:** Anomalous geopotential height (shading, 10 gpm) at 500 hPa in November and December 2015 and its stationary wave activity flux (vector, $m^{-2} s^{-2}$) (An et al., 2020).

*6.  Why show visibility in Figure 2, but PM5 concentrations in Figure 15? Can you kindly show me both of the anomalies visibility and PM2.5 concentrations in the reply letter (relative to climatology), associated with SR−NH, SR−noNH and the left samples?*

**Response:** Thank you very much. In this study, PM$_{2.5}$ concentration was considered as an assisting data to investigate the distribution of particulate matter. It does not change the main conclusions in this manuscript. According to your suggestion, we show PM2.5 anomaly (relative to climatology of 1985−2015) associated with SR−NH and SR−noNH (Fig. 3S). Since we take the visibility of 10 km as the threshold for screening haze cases, its anomaly might not be a good indicator for our research, so we show the real value of visibility in Fig. 4S. From Fig. 3S, PM$_{2.5}$ concentration in the NCP is positive anomaly in SR−NH events, while PM$_{2.5}$ concentration in the NCP is negative anomaly in SR−noNH events. In SR−NH events, visibility in most areas of the NCP is significantly lower than

10km, while in SR−noNH events, almost no visibility in the NCP is lower than 10km. Therefore, visibility and PM$_{2.5}$ concentration can represent the atmospheric pollution in the NCP in this study.

[Figure]

**Figure 3S:** Composite maps of PM$_{2.5}$ concentration anomaly for (a) SR−NH events, (b) SR−noNH events, and (d) difference between SR−NH events and SR−noNH events. White dotted regions indicate areas at the 95% confidence level based on the two-tailed Student's t test.

[Figure]

**Figure 4S:** Composite maps of visibility for (a) SR−NH events, (b) SR−noNH events, and (d) difference between SR−NH events and SR−noNH events.

*Specific comments:*

1. *Abstract: Some abbreviations, that occurred only once (or less than 3 times), are not necessary in abstract. Too much abbreviations are not easy to read. Possibly, the authors could also check the main texts throughout.*

**Response:** Thanks for your suggestion. Yes, we also realize that a large number of abbreviations might cause confusion to readers. Therefore, we have cancelled some abbreviations in the manuscript.

**Line 19:** "… with the north−south circulation system,"

**Line 23:** "… A linear baroclinic model,"

**Line 24:** "… observed north−south circulation system …"

**Line 24−25:** "Quasi-geostrophic vertical pressure velocity diagnostics …"

**Line 25:** "… the north−south circulation system made …"

**Line 26−27:** "The results indicated that the north−south circulation system …"

**Line 53:** "the SST anomalies …"

**Line 53−55:** "the Pacific Decadal Oscillation (Zhao et al., 2016); the Arctic Oscillation (Cai et al., 2017; G. Zhang et al., 2019); the preceding Antarctic oscillation (i.e., August−September−October) (Z. Zhang et al., 2019); and the North Atlantic Oscillation (Feng et al., 2019; Li et al., 2021),"

**Line 63:** "… the local north−south circulation system (Fig. 1)."

**Line 79:** "… from Chinese Meteorological Administration (http://www.nmic.cn/) …"

**Line 81:** "… from Chinese Meteorological Administration …"

**Line 94:** "… by Chinese Meteorological Administration (2010),"

**Line 96−97:** "… by Chinese Meteorological Administration …"

**Line 124:** "… The LBM (Watanabe and Kimoto, 2000),"

**Line 154−155:** "but they also weaken the East Asian winter monsoon (An et al., 2020),"

**Line 165:** "… the Eurasian teleconnection (EU),"

**Line 167:** "… the development of the East Asian winter monsoon (An et al., 2020),"

**Line 195:** "… forms the local north−south circulation system …"

**Line 217:** "… the local north−south circulation system seems …"

**Line 226:** "The local north−south circulation system simulated by …"

**Line 227:** "of the local north−south circulation system (Figs 5 and 10),"

**Line 227:** "… the results obtained by An et al. (2020)."

**Line 251:** "which in turn maintains the local north−south circulation system…"

**Line 259:** "… (referred to as the local north−south circulation system in this paper)."

**Line 266:** "… a similar the local north−south circulation system (not shown),"

**Line 282−283:** "the Arctic Oscillation (Cai et al., 2017),"

**Line 286−287:** "… the China Meteorological Administration (http://data.cma.cn/, 2017),"

**Line 436:** "… of the local north−south circulation system."

*2. Line 26: The 'diabatic heating' was defined as Q much later (Line 107).*

**Response:** Thank you very much. We have revised the sentence in the manuscript.

**Line 26:** "… and diabatic heating (Q)."

*3. Line 30: The range of NCP should be added in the text, and the others are similar.*

**Response:** Thanks for your suggestion. We have added the range of the NCP in the manuscript.

**Line 13−14:** "… the North China Plain (NCP, 30−40.5°N, 112−121.5°E) suffered many periods of heavy haze, …"

**Line 31:** "Extensive heavy haze on the North China Plain (NCP, 30−40.5°N, 112−121.5°E) has …"

*4. Line 46−47: "Overall, the role of meteorology in the generation of haze is crucial but uncertain, and may be closely related to the regulation of the large-scale circulation". This sentence must be rephrased, because the meanings are confused.*

**Response:** We are sorry for this confusion. We have rephrased this sentence in the manuscript.

**Line 47:** "Overall, the role of meteorology modulated by the large-scale circulation in the generation of haze is crucial but uncertain."

*5. Line 51: The abbreviation of SSTAs may be not necessary.*

**Line 51:** "the SST anomalies related to …"

215  *6.  Line 83: PM2.5 here has a format error.*

**Line 85:** "These daily PM$_{2.5}$ concentrations …"

*7.  Line 83−85: Because the climate scientist must carefully choose the data series, can you show the readers about the quality assessment of Yang's PM5 datasets for the period 1980–2019? In my opinion, excellent agreement with ground measurements during 2013–2019 did not illustrate good performance for the period 1980–2019.*

**Response:** Yes, you are right. We also realize that there might be some limitations in using this dataset to investigate atmospheric pollution process on the synoptic scale. According to Li et al. (2021), PM$_{2.5}$ dataset is only assessed based on observation data from 2014 to 2019 due to the lack of long-term PM$_{2.5}$ data. The dataset is obtained by machine learning based on supplementary data such as visibility observation data from 1980 to 2019. A continuous increase of the mean PM$_{2.5}$ from 1985 to 2014 shows a similar trend to PM$_{2.5}$ data simulated with GEOS-Chem model by Yang et al. (2016) (Li et al., 2021) (Fig. 2S and Fig. 3S). It might be a good choice to study the interannual variation of air pollution in the NCP based on this dataset. For the weather scale pollution process, however, it might have some limitations. Therefore, it was only considered as an assisting data for this study. It will not change main conclusions in the manuscript.

[Figure]

**Figure 5S:** Time series of annual mean (purple lines) and seasonal means of modeled PM2.5 (colored solid lines) from 1980 to 2019 and corresponding observed PM2.5 (colored dashed lines) from 2015 to 2019 averaged over China and the NCP (Li et al., 2021).

[Figure]

**Figure 6S:** The time series of (a) simulated surface-layer PM2.5 concentrations from the CTRL simulation ($\mu g\,m^{-3}$), observed haze days (days), and MODIS aerosol optical depth (AOD) averaged over 196 stations in eastern China in DJF for 1985–2005, 1980–2014, and 2000–2013, respectively (Yang et al., 2016).

*8. Line 162: Add the full name of 'EU' if it is the first time it appears.*

**Line 165:** "… the Eurasian teleconnection (EU),"

*9. Line 251: "the appearance of haze over the NCP" may only be some of haze event, which related to the rainfall over southern China.*

**Response:** Thanks for your suggestion. We have revised the sentence in the manuscript.

**Line 255:** "… the appearance of some of haze events over the NCP …"

*10. Figure 1: the color of the circular arrow was confusing.*

**Response:** We are sorry for this confusion. We have replotted Figure 1 as your suggestion.

[Figure]

**Figure 1:** Schematic illustration of the local north−south circulation system. The red (blue) circular arrow indicates the anomalous cyclone (anticyclone), the red (blue) vertical arrow indicates ascending (descending) motion, the thick black arrow indicates northward movement in the upper-level troposphere, H indicates the anticyclonic anomaly, and the white cloud indicates rainfall (see An et al. (2020) for caption details).

*11. Figure 6: ω>0 (Pa s–1, shading) means descending motion, why is it an ascending motion here?*
*Is it multiplied by −1?*

**Response:** We are sorry for this confusion. Yes, ω was multiplied −1 as you guessed. We have added more detail to the caption of Figure 6.

**Line 463−466:** Figure 6: Composite sections of absolute values for the 13 SR–NH events: latitude–height sections of (a) vertical velocity (shading, unit: –1 Pa s⁻¹) and wind vectors (v and –ω) averaged over 112°–120° E, and longitude–height cross sections of (b) vertical velocity (shading, unit: –1 Pa s⁻¹) and wind vectors (u and –ω) averaged over 20°–30° N and (c) 30°–40° N. White dotted regions indicate areas at the 99% confidence level based on the two-tailed Student's t test.

*12. Figure 7: The positive velocity potential may represent divergence.*

270     **Response:** Yes, you are right. According to the definition of the velocity potential $\varphi$ by Tanaka et al. (2004),

$$D = \nabla \cdot V = -\nabla^2 \varphi,$$

here, the positive velocity potential represents divergence. The divergent wind $V\varphi$ flows from the maximum to the minimum of the velocity potential field (Tanaka et al., 2004).

275     It is noted that the *Helmholtz* theorem and the original definition by Lamb (1945) has no minus sign on the right-hand side, which implies that the negative velocity potential represents divergence. In this study, we calculated the velocity potential following the definition of $D = \nabla^2 \varphi$. The negative velocity potential represents divergence.

As Fig. 7S, Weng et al. (2007) also calculated velocity potential according to the definition of $D = \nabla^2 \varphi$. We have added the illustration for the velocity potential shown in Fig. 7.

280

**Line 471:** "The divergent wind flows from the minimum to the maximum of the velocity potential field."

[Figure]

**Figure 7S:** Partial correlation patterns of the 200 hPa divergent winds (arrow) and velocity potential anomalies (shading) with a) EMI and b) Niño3 in the domain of (70°E–40°W, 80°S–80°N). The wind vectors with correlation coefficients of either zonal or meridional component that are not significant at the 90% level are omitted (Weng et al., 2007).

*13. Figure 10: "As Fig. 5," may be "Fig. 6"?*

**Response:** Yes, it is Fig. 6. We have revised the caption of Figure 10.

**Line 484:** Figure 10: As Fig. 6, except for the results from the LBM. For clarity, ω is multiplied by −20.

*14. How did you composite the maps associated with 13 SR−NH events. I did not find it in any of the Figure captions.*

295 **Response:** We are sorry that we didn't add an illustration for composite maps. In this study, we firstly calculated average all days of each case, then all case were averaged for composite the map. We have added the description of composite analysis in section 2.

**Line 131:** "All composite maps were obtained from the average of each individual case."

[revised manuscript text omitted]

---

## Author Comment (AC2)

Dear reviewer #2,

We appreciate you for carefully reviewing our manuscript and providing the valuable suggestions to improve our paper. We have carefully read all comments and revised the manuscript as suggested. The following are our responses to all comments point by point. The italicized sentences are all comments, and the other sentences are the author's responses. The blue sentences and words are the specific revisions. We also marked all relevant changes in the manuscript in the same way.

*The manuscript by An et al. investigates the effects of rainfall-related diabatic heating over southern China on the wintertime haze events over northern China plain (NCP). The authors suggest that the NCP haze event is modulated by the Rossby wave train emanating from the North Atlantic and the secondary circulation induced by the heavy rainfall over southern China. Specifically, the authors argued that the diabatic heating associated with the heavy rainfall over southern China leads to descending motions over NCP, which reinforce the anticyclonic anomaly produced by the Rossby wave train and thus favor the formation of haze events.*

*Overall, the flow of the paper and the figures used to support the arguments are cohesive. However, I am not fully convinced about the role of diabatic heating over southern China in the haze events over NCP. I recommend that the paper be considered for publication after addressing the major comments below.*

*Major comments:*

1. *The authors found that the NCP haze is modulated by the Rossby wave train emanating from the North Atlantic and the secondary circulation induced by the heavy rainfall over southern China. The diabatic heating associated with the heavy rainfall over southern China leads to descending motions over NCP, which reinforce the anticyclone resulting from the Rossby wave train. I am*

*not fully convinced by this argument because (1) in observations, it is difficult to disentangle the effect of diabatic heating over southern China on the anticyclone from the Rossby wave train; (2) the LBM simulation doesn't reproduce the observed anticyclonic anomaly over NCP (compare Fig. 9a with Fig. 4b; Fig. 11).*

**Response:** Thank you very much. We are sorry for this confusion. As you said, it is difficult to find the connection between the anticyclone and diabatic heating over southern China. Therefore, we highlight the role of diabatic heating over southern China on the local north-south vertical circulation over eastern China (Fig. 1 and Fig. 6). This local north-south vertical circulation is directly related to the ascending motion, which can be strengthened by adiabatic heating released by rainfall in southern China. To verify the effect of adiabatic on the local north-south vertical circulation in observations, we projected a numerical experiment based on the LBM model. From results of the LBM model, there is a similar pattern of the local north-south vertical circulation to the observations (Fig. 10 and Fig. 11). Moreover, we calculated the components of the ω equation as described in Eq. 4 of the manuscript. In Fig. 12 of the manuscript, compared with dry dynamic forcing term, the adiabatic heating term can reproduce pattern that is closer to vertical velocity, which means that adiabatic heating does enhance the local north−south vertical circulation. In addition, from current results in the manuscript, the LBM simulation doesn't reproduce the observed anticyclonic anomaly in North China. On the one hand, it might be related to the simple dynamic framework of the LBM model (Watanabe and Kimoto, 2000). On the other hand, it might be that adiabatic heating cannot excite the anticyclone in North China. As we all known, there seems to be no classical theory to support the opposite of this second guess. Overall, this is a good topic needed to be further studied in the future. In current study, we mainly emphasize that adiabatic heating affects the local north−south vertical circulation, which weakens the vertical diffusion of particulate matter in the NCP. In order to further verify this conclusion, we show the distribution of the boundary layer height anomaly in SR−NH (Fig. 8Sa), SR−noNH (Fig. 8Sb) and difference between SR−NH and SR−noNH. The boundary layer height anomaly of SR−NH events in the NCP is significantly lower than that in SR−noNH events, which is not conducive to the vertical diffusion of

particulate matter in the NCP (Fig. 8S). This supports our results in the manuscript. We have added the information from Fig. 8S in the manuscript.

[Figure]

**Figure 8S:** Composite maps of the boundary layer height anomaly for (a) SR−NH events, (b) SR−noNH events, and (d) difference between SR−NH events and SR−noNH events. White dotted regions indicate areas at the 95% confidence level based on the two-tailed Student's t test.

**Line 231:** "… upper-level convergence and low-level weak divergence over the NCP …"

**Line 251−252:** "… leading to a shallow atmospheric boundary layer height, finally aggravates the haze pollution over the NCP."

2. *L131: Could the authors elaborate on the definition of extreme winter rainfall events? Is the definition based on monthly rainfall or daily rainfall averaged over southern China? Why there are 22 rainfall events during 1985−2015 and why the durations of each individual event are different (Tables 1 and 2)?*

**Response:** We are sorry for this confusion. Persistent rainfall events were defined as follows (Ding and Li, 2017):

First, rainfall process should continue 3 days or longer. Second, daily maximum rainfall should exceed 50 mm at least 1 day. Third, more than two provinces were influenced by rainfall process. Fourth, the maximum precipitation of rainfall process during the whole event should exceed 100 mm. Lastly, of the interval between two persistent heavy rainfall process was less than 3 days, the two events were marked as one event. The definition is based on daily rainfall averaged over

southern China. Daily rainfall data were international exchange 194-stations observed data, taken from China Meteorological data sharing service system, Chinese Meteorological Administration. According to the five strict steps, 19 persistent rainfall events during 1985−2015 were selected. Because rainfall events are persistent rainfall, the durations of each individual event are different.

3.  *L137: When heavy rainfall fell over southern China, the probability for haze to occur over NCP is ~59% (13 out of 22 extreme rainfall events). Although the authors have compared the atmospheric circulations between SR−NH (13 events) and SR−noNH (9 events) events, I feel that the upper-tropospheric Rossby wave trains look very similar (c.f., Fig. 4a and Fig. 14a). Instead, the significance of the Rossby wave train reduces in the SR−noNH events, which might suggest more variabilities in the wave train. Could the authors explicitly show the differences between Fig. 4 and 14?*

**Response:** Thank you very much. Before replying this question, we have to correct an error in our manuscript. After our check, we found that the proportion of SR−NH is 68.42%. 59.9% was obtained in the original because we counted three more SR−noNH events. The new proportion (68.42%) does not change our current research results. We have corrected this error in the latest manuscript. We are sorry for this mistake.

Yes, you are right. The two Rossby wave trains are the background circulation of SR−NH events. We have confirmed the role of the Rossby wave trains on SR−NH events in section 3 of the manuscript. In the mid and upper troposphere, the differences between these two weathers seem to be weak except that the 500-hPa northeast Asian anomalous anticyclone is more westerly and the subtropical westerly jet wave train seems to be stronger in SR−NH events (Fig. 4 and Fig. 14). However, from Fig. 8S, the atmospheric boundary layer height in SR−NH events is more shallow than that in SR−noNH events. The local north−south vertical circulation can restrain the development of the atmospheric boundary layer height through descending motion in the NCP. This means that rainfall in southern China can affect haze in the NCP.

4. *While the authors have compared the atmospheric circulations during SR−NH events and SR−noNH events, how many NH events occur without SR? Is the atmospheric circulation during NH−noSR events also controlled by Rossby wave train similar to the one shown in Fig, 4b? This might help illustrate the importance of SR rainfall in observations.*

**Response:** Thanks for your suggestion. Your suggestion is very good. However, as we listed in the introduction of the manuscript, the previous have found that there are many factors affecting haze in North China, and the mechanism is very complex, and there might be interaction between different factors (Yin et al., 2021). In addition to factors listed in introduction of the manuscript and rainfall in southern China, the Tibetan Plateau topography (Xu et al., 2016), stratospheric polar vortex (Huang et al., 2021), and Siberian High (Zhao et al., 2018) also plays an important role in haze in the NCP. If we do not limit a common background (such as rainfall and haze), there are also many haze events, and the formation mechanism might be different. The Rossby wave trains in this manuscript and rainfall in southern China is one of the potential factors affecting haze in the NCP (68.42% of this study). However, we could not attribute the haze events in the NCP to rainfall in southern China or the Rossby wave trains. The causes of haze in North China need to be studied more comprehensively in the future.

*Technical corrections:*

1. *L51−56: I wouldn't call the Eurasian snow cover, ENSO, and Arctic sea ice changes as atmospheric conditions*

**Response:** Thank you very much. We have revised the sentence in the manuscript.

**Line 50:** "… wintertime heavy haze over the NCP stems from the underlying meteorological factors."

2. *L108: add a space between QG and w*

**Response:** Thanks for your suggestion. We have added a space between QG and *w*.

**Line 108:** "The QG ω equation reads:"

3. *L222: "The NSC simulated by the LBM bears a striking resemblance to the observed spatial pattern of the NSC (Figs. 5 and 10)". Should "Figs. 5" be "Fig. 6"? Also, for the caption of Fig. 10, should be "as Fig. 6" instead of "as Fig. 5".*

**Response:** Thanks for your suggestion. We have revised the sentence as your suggestion.

**Line 227:** "… (Figs 6 and 10) …"

**Line 484:** "Figure 10: As Fig. 6, …"

4. *L222: Given the substantial differences between Figs. 6 and 10, I personally wouldn't say the NSC simulated by LBM bears a striking resemblance to observation.*

**Response:** Thank you very much. We have revised the expressions like this in the manuscript.

**Line 226:** "… by the LBM is similar to the observed spatial pattern …"

5. *L205: Fig. 9a should be Fig. 8a*

**Line 209:** "… (Fig. 8a)."

6. *L208: Fig. 9b should be Fig. 8b*

**Line 212:** "… (Fig. 8b)."

7. *L216: Fig. 10a, may be Fig. 9a*

**Line 220:** "… (Fig. 9a)."

8. *L218: Fig. 8b, may be Fig. 9b*

**Line 222:** "… (Fig. 9b)."

9. *Fig. 12: the first and second rows show the diabatic heating and dynamic forcing terms, of which the sign is opposite to the omega. For ease of comparison with the omega shown in the bottom row, I suggest reversing the sign of diabatic heating and dry forcing terms.*

**Response:** Thanks for your suggestion. We have reversed the sign of diabatic heating and dry forcing terms shown in Fig. 12 according to your suggestion.

[Figure]

**Figure 12:** Composite QG decomposition for the 13 SR−NH events. Each column shows a different level. From top to bottom, the rows show the diabatic heating term ($\frac{R}{pf^2}\nabla^2 Q$), the dry forcing term ($F = \frac{1}{f}\partial_p Adv_\varsigma + \frac{R}{pf^2}\nabla^2 Adv_T$), where $Q$ is calculated using the Eq. 1), and $\omega$ (reanalysis data),

respectively. The unit of $\omega$ is Pa s$^{-1}$. The unit of the $F$ and Q terms is Pa$^{-1}$ s$^{-1}$. For ease of viewing, and to reduce the differences in scale, the $F$, Q terms, and $\omega$ were multiplied by $4 \times 10^{7}$, $0.8 \times 10^{7}$, and 4, respectively. White dotted regions indicate areas at the 95% confidence level based on the two-tailed Student's t test.

*10. Is the omega shown in Fig. 6 also multiplied by −20 as that in Fig. 10? If not, the magnitude of omega is significantly different in the LBM model and observation.*

**Response:** We are sorry for this confusion. The omega shown in Fig. 6 is not multiplied by −20. As we all know, the LBM is an idealized linear baroclinic model to understanding the complicated sequence of feedback in the dynamic atmosphere, by removing nonlinearity in their processes. The dynamical framework is simplified in this model, so that the results would be much easily interpreted (Watanabe and Kimoto, 2000). However, compared to the actual atmosphere affected by many other factors, all the anomalous atmospheric circulation in the model are responses for the prescribed heating source. The setting of friction and damping coefficient also effects the magnitude of anomalous circulation response to a certain extent (Watanabe and Kimoto, 2000). In addition, there is a positive feedback process between the upward motion in southern China and anomalous precipitation, making the anomalous upward motion increase continuously, which do not exist in the LBM model. Therefore, the results from the LBM might be smaller than observation, but this does not affect the qualitative understanding of the physical mechanism. In this study, the LBM model was aimed to qualitatively understand that the adiabatic heating related to rainfall in southern China can really product a local north−south vertical circulation over eastern China.

*11. L47−49: "Large-scale circulation, and the related external forces derived via exciting the teleconnection pattern, regulate meteorological conditions, reduce dispersion, and facilitate the accumulation of haze pollutants (Zhang et al., 2020)." Please consider rephrasing this sentence.*

**Response:** Thanks for your suggestion. We have rephrased this sentence in the manuscript.

**Line 48−49:** "Large-scale circulations regulate meteorological condition liking reducing dispersion by atmospheric teleconnection and facilitate the accumulation of haze pollutants (Y. Zhang et al., 2020)."

[revised manuscript text omitted]

---

## Author Response (AR2)

Dear editor,

We deeply appreciate you and two reviewers for carefully reviewing our manuscript and providing the valuable suggestions to improve our paper. We have carefully read your comments and revised the manuscript as suggested. The red words and sentences in the manuscript are the specific revisions for your suggestion.

Thank you and best regards.
Yours sincerely,
Lifang sheng